# aPKC and F-actin dynamics promote Hippo pathway polarity in asymmetrically dividing neuroblasts

**Niranjan S. Joshi[1],*, Victoria M. Sullivan[1], Sherzod A. Tokamov[2,3,‡] and Richard G. Fehon[2,3,§]**

## ABSTRACT

The Hippo signaling pathway is conventionally known to restrict tissue growth in animals. Genetic studies have also shown that loss of Hippo pathway components leads to defects in asymmetric cell division in *Drosophila* neural stem cells, known as neuroblasts. The hallmark of neuroblast division is the asymmetric localization of aPKC/Bazooka (Par-3)/Par-6 complex, termed the Par complex, to the apical cell cortex. However, the localization of the Hippo pathway components in neuroblasts remains unknown. Here, we report that two key activators of the Hippo pathway, Kibra and Salvador, polarize to the apical cortex of mitotic neuroblasts. We show that apical polarity, via the activity of aPKC, and F-actin dynamics synergize to drive Kibra polarization. Together, these results provide further insights into the relationship between apical polarity and Hippo pathway organization and suggest a possible mechanism by which pathway activity is regulated during neuroblast asymmetric division.

KEY WORDS: Kibra, aPKC, Hippo, Mitosis, Polarity, Cytoskeleton

## INTRODUCTION

The Hippo pathway is an evolutionarily conserved regulator of tissue growth (Misra and Irvine, 2018; Pan, 2022). When active, the Hippo pathway's core kinase cassette phosphorylates the transcriptional coactivator Yorkie (Yki), excluding it from the nucleus and repressing growth. The core kinase cassette includes the serine/threonine kinases Tao, Hippo (Hpo) and Warts (Wts), joined by the scaffold proteins Salvador (Sav) and Mob as tumor suppressor (Mats). In the absence of phosphorylation, Yki enters the nucleus and drives transcription of pro-growth genes. The assembly of the core kinase cassette is promoted by upstream components including Kibra (Kib), Merlin (Mer), Crumbs and Expanded (Ex). Because the pathway was discovered in a series of mutant screens looking for overgrowth (Pan, 2022), subsequent research has largely focused on its role in growth control via repression of Yki activity, leaving

questions about its role in other cellular and developmental contexts unanswered.

One context in which the Hippo pathway plays an important role unrelated to growth control is in the asymmetric cell division of *Drosophila* neural stem cells ('neuroblasts'). During embryonic and larval development, neuroblasts divide asymmetrically, resulting in one daughter that renews the neuroblast identity and another that eventually differentiates into a neuron. Studies in neuroblasts have generated much of what is known about the mechanisms that underlie asymmetric cell division, a common biological strategy used to produce cellular diversity (Loyer and Januschke, 2020; Sunchu and Cabernard, 2020). In neuroblasts, asymmetry is driven by evolutionarily conserved proteins aPKC, Par-3 (Bazooka or Baz, in *Drosophila*), and Par-6, which collectively form the 'Par complex' (Fig. 1A). The Par complex polarizes to the apical cortex at the onset of mitosis and directly excludes pro-neuronal proteins, causing them to relocalize to the basal pole (Smith et al., 2007; Atwood and Prehoda, 2009; Homem and Knoblich, 2012). Simultaneously, the Par complex interacts with the mitotic spindle via a variety of cofactors – including Partner of Inscuteable (Pins), Discs large (Dlg), Canoe (Cno), Mushroom body defective (Mud), and Kinesin heavy chain 73 (Khc73) – to orient the mitotic spindle in the apical-basal axis and ensure proper segregation of pro-neural fate determinants (Siegrist and Doe, 2005; Speicher et al., 2008; Loyer and Januschke, 2020).

The Hippo pathway promotes both apical cortical polarity and proper spindle alignment in dividing neuroblasts. Animals trans-heterozygous for mutations in apical polarity and Hippo pathway components – including *wts*, *hpo*, *sav*, *mats*, *mer*, and *ex* – display ectopic neurons, a hallmark of aberrant neuroblast polarity (Keder et al., 2015). Additionally, neuroblasts lacking *sav*, *hpo*, or *wts* fail to properly polarize the Par complex and align the mitotic spindle in the apical-basal axis (Keder et al., 2015). Depletion of Sav, Hpo, or Wts also causes spindle misorientation in an induced polarity system in *Drosophila* S2 cells (Dewey et al., 2015). These phenotypes are independent of Yki activity but require pathway activity (Dewey et al., 2015; Keder et al., 2015). Active Wts phosphorylates Cno and Mud, which promotes their association with Khc73 and Pins (Dewey et al., 2015; Keder et al., 2015). Wts also phosphorylates Baz, but the significance of this interaction remains unknown (Keder et al., 2015).

Little is known about the mechanism underlying Hippo pathway activation in neuroblasts. Here, we combine live imaging with genetic and chemical manipulations to examine the dynamic behavior of key upstream Hippo pathway activators – including Kib, Mer, Sav, and Wts – in living neuroblasts. We find that Kib and Sav polarize cyclically to the apical pole in dividing neuroblasts. Furthermore, we demonstrate that aPKC activity and F-actin dynamics synergize to promote Kib's apical location. Together, these results provide mechanistic insight into how upstream Hippo pathway components are apically localized and suggest a mechanism for polarized pathway activation in the context of asymmetric cell division.

[1]The College, The University of Chicago, Chicago, IL 60637, USA. [2]Department of Molecular Genetics and Cell Biology, The University of Chicago, Chicago, IL 60637, USA. [3]Committee on Development, Regeneration, and Stem Cell Biology, The University of Chicago, Chicago, IL 60637, USA.
*Present address: Department of Cell Biology, School of Medicine, Duke University, Durham, NC 27708, USA. ‡Present address: Howard Hughes Medical Institute, Department of Plant and Microbial Biology, University of California, Berkeley, Berkeley, CA 94720, USA.

§Author for correspondence (rfehon@uchicago.edu)

R.G.F., 0000-0003-4889-2602

## RESULTS

### The Hippo pathway activator Kib polarizes in neuroblasts

We began by characterizing the behavior of the Hippo pathway activator Kib in live neuroblasts. Several pieces of evidence suggest that it may play a role in organizing Hippo signaling in neuroblasts. First, it is known to be a key upstream activator of the Hippo pathway components Sav, Hpo, and Wts (Hamaratoglu et al., 2006; Baumgartner et al., 2010; Genevet et al., 2010; Yu et al., 2010; Su et al., 2017). Second, *Mer*, the product of which binds and synergizes with Kib in imaginal and glial cells (Baumgartner et al., 2010; Reddy and Irvine, 2011; Su et al., 2017), has one of the most penetrant genetic interactions with *cno* of all genes involved in the Hippo pathway (Keder et al., 2015). Third, Kib is known to be regulated by cell polarity, which it also regulates in turn (Yoshihama et al., 2011, 2012; Jin et al., 2015; Tokamov et al., 2023), and cytoskeletal dynamics (Tokamov et al., 2023), both of which are important for neuroblast asymmetry (Oon and Prehoda, 2019).

To observe Kib's behavior in neuroblasts, we imaged intact neuroblasts in living larval brain explants expressing GFP- or Halo-tagged versions of Kib under the *ubiquitin (ubi)* promoter (Ubi>Kib-GFP or Ubi>Kib-Halo), as we have done previously (Tokamov et al., 2021, 2023). Strikingly, we observed that Ubi>Kib-GFP polarizes in concert with the cell cycle (Fig. 1B; Movie 1). Ubi>Kib-GFP was diffusely cytoplasmic prior to mitotic entry but subsequently formed a strong apical crescent during prophase. While the crescent persisted until telophase, its intensity diminished after nuclear envelope breakdown (NEB), indicating that Kib polarization is transient. We confirmed that polarization was not an artefact of Kib overexpression by observing endogenously expressed Kib::YFP in larval neuroblasts, which also formed polarized crescents (Fig. S1A). Using Airyscan microscopy for better spatial resolution, we found that the Ubi>Kib-GFP apical crescent was not contiguous but rather a conglomerate of smaller Ubi>Kib-GFP puncta (Fig. 1C; Movie 2).

To characterize the localization of Kib relative to known apical polarity proteins, we coexpressed Ubi>Kib-GFP or Ubi>Kib-Halo with the apical proteins Baz-GFP, aPKC-Halo, and Dlg-GFP. Consistent with prior work (Oon and Prehoda, 2019), Baz-GFP and aPKC-Halo appeared diffusely cytoplasmic prior to mitotic entry but formed apical crescents during prophase (Fig. 2A,B; Movies 3 and 4). These crescents became brighter after NEB and during metaphase,

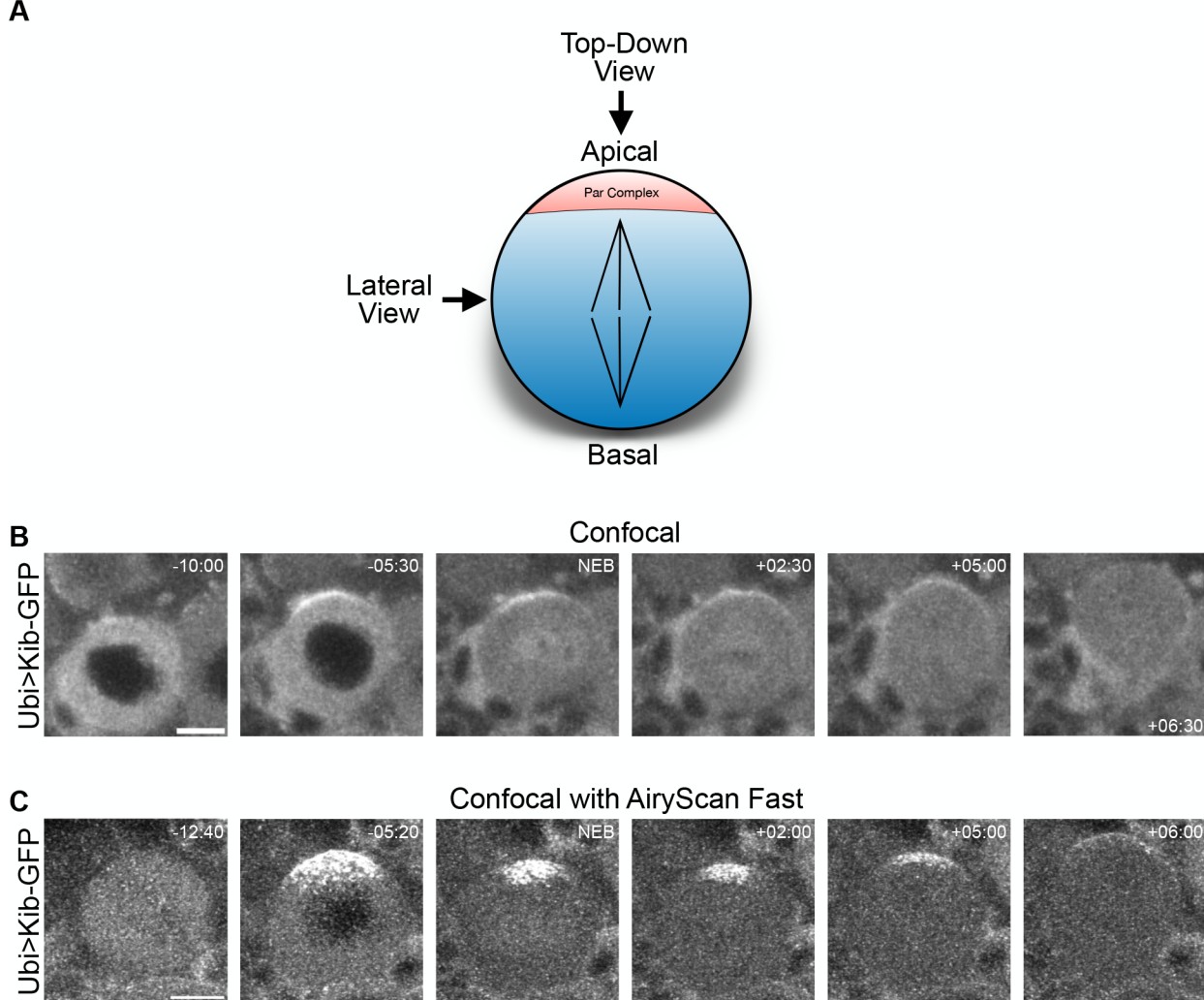

**Fig. 1. Kibra polarizes to the apical cortex in living neuroblasts.** (A) Cartoon of a mitotic neuroblast at metaphase. The Par complex and associated apical proteins form apical crescents, depicted in red. The spindle is oriented parallel to the apical-basal axis, shown in black. Unless otherwise noted, all images are taken from the lateral view and shown with apical up and basal down. (B) A mitotic neuroblast expressing Ubi>Kib-GFP. Times indicated here and hereafter are relative to NEB. (C) A mitotic neuroblast expressing Ubi>Kib-GFP imaged using Airyscan super-resolution, showing that Kib is punctate at the apical pole. Scale bars: 5 μm.

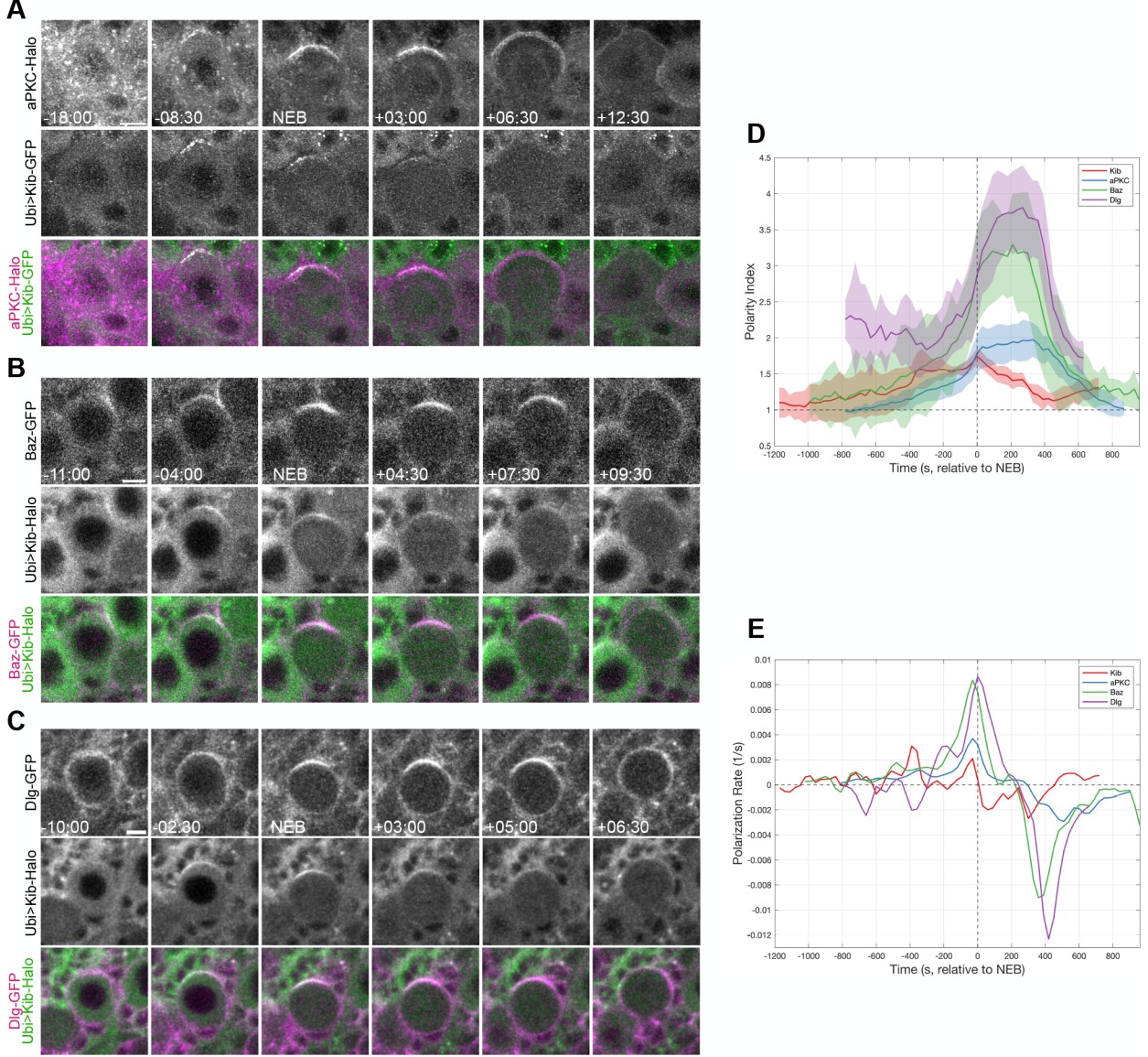

**Fig. 2. Kibra polarizes in a manner similar to known polarity proteins.** (A) A mitotic neuroblast expressing endogenous aPKC-Halo (Erdmann et al., 2019) and Ubi>Kib-GFP. (B) A mitotic neuroblast expressing Baz-GFP and Ubi>Kib-Halo. *baz* was tagged at its endogenous locus (Buszczak et al., 2007). (C) A mitotic neuroblast expressing Dlg-GFP and Ubi>Kib-Halo. *dlg* was tagged at its endogenous locus (Buszczak et al., 2007). (D) The polarity indices of neuroblasts expressing aPKC-Halo, Baz-GFP, Dlg-GFP, or Ubi>Kib-GFP plotted as a function of time relative to NEB. Eight to 12 total neuroblasts are represented in each curve, and each timepoint represents the average of at least three neuroblasts. Shading represents one standard deviation from the mean. The dotted vertical line marks the time=NEB, and the dotted horizontal line marks the polarity index=1. (E) The derivative of each curve in D. Scale bars: 5 µm.

before diminishing during subsequent stages of the cell cycle. Dlg-GFP appeared in the cytoplasm but was symmetrically enriched around the cortex prior to late prophase (Fig. 2C; Movie 5). It became apically polarized around NEB, consistent with the fact that it is thought to promote spindle alignment at metaphase (Siegrist and Doe, 2005), before depolarizing in anaphase. Kib appeared to colocalize at the apical cortex with polarized Baz-GFP and aPKC-Halo from prophase to anaphase (Fig. 2A,B) and with polarized Dlg-GFP after NEB (Fig. 2C).

We next characterized the dynamics of Kib polarization relative to that of known apical polarity proteins. To do so, we imaged live neuroblasts expressing either Ubi>Kib-GFP, Baz-GFP, aPKC-Halo, or Dlg-GFP and quantified the ratio between the mean fluorescence intensity of the apical cortex and the basal cytoplasm as a measure of

the extent of polarization ('polarity index'). We plotted the average polarity index of multiple neuroblasts over time for each protein (Fig. 2D) and the derivative of each polarization curve to show the dynamics of the rate of polarization (Fig. 2E). Interestingly, we found that Ubi>Kib-GFP, Baz-GFP, and aPKC-Halo accumulated apically at similar rates from early prophase until ~400 s before NEB. At that point, the rate of Ubi>Kib-GFP polarization remained approximately constant while that of Baz-GFP and aPKC-Halo polarization increased dramatically. After NEB, Ubi>Kib-GFP immediately began to depolarize, while Baz-GFP and aPKC-Halo began to depolarize ~300 s later. Dlg-GFP began to polarize only ~300 s before NEB, when Ubi>Kib-GFP was already strongly polarized. Dlg-GFP continued to polarize after NEB, while Ubi>Kib-GFP was depolarizing and depolarized at approximately

the same time as Baz-GFP and aPKC-Halo. These observations indicate that Kib is recruited to the apical cortex independently of Dlg and suggest that aPKC and/or Baz may play a role in Kib's polarization.

## aPKC's kinase activity is required for Kib polarization

We next tried to uncover how Kib polarization is regulated. Kib and its mammalian ortholog WWC1/KIBRA physically interact with aPKC and its mammalian ortholog, PKCζ (Yoshihama et al., 2011; Yoshihama et al., 2012; Jin et al., 2015; Tokamov et al., 2023). Biochemical studies of WWC1/KIBRA have shown that it can be phosphorylated by PKCζ (Büther et al., 2004), and both *Drosophila* Kib and mammalian WWC1/KIBRA have a well-conserved consensus aPKC substrate site (Soriano et al., 2016). Furthermore, this domain is necessary and sufficient for recruitment of Kib by aPKC in *Drosophila* S2 cells (Tokamov et al., 2023). These observations suggest that aPKC might control Kib polarization in neuroblasts.

To ask if aPKC is necessary for Kib polarization, we observed the effect of inhibition of aPKC kinase activity on Ubi>Kib-GFP localization. Genetic loss of aPKC has pleiotropic effects on neuroblast polarity, including disruption of Par-6, Lgl, and Mira localization (Rolls et al., 2003). To minimize these effects, we used

$aPKC^{as4}$, an aPKC allele that encodes a functional protein that can be acutely inhibited by the cell-permeable small molecule 1-NA-PP1 (Hannaford et al., 2019). We exposed brain explants to 1-NA-PP1 for 15 min, which is sufficient to inhibit aPKC in larval neuroblasts and imaginal discs (Hannaford et al., 2019; Tokamov et al., 2023). Ubi>Kib-GFP polarized normally in dividing neuroblasts when brains were pretreated with DMSO, but Ubi>Kib-GFP apical crescents failed to form when brains were pretreated with 1-NA-PP1 (Fig. 3A,B; Movie 6). Additionally, wild-type brains treated with 1-NA-PP1 displayed normal apical Kib polarity (Fig. 3B), indicating that effects we observed using the $aPKC^{as4}$ allele were due to loss of aPKC activity. Thus, aPKC activity is required for Kib polarization.

## aPKC does not physically tether Kib to the apical cortex

Acute inhibition of $aPKC^{as4}$ by 1-NA-PP1 results in a uniform cortical distribution of aPKC in larval neuroblasts (Hannaford et al., 2019). Thus, our observation that 1-NA-PP1 disrupts Kib polarization is consistent with the possibility that aPKC recruits Kib through physical interaction. Alternatively, aPKC kinase activity could control Kib localization through another mechanism that does not involve physical interaction. To further test the possibility that aPKC physically recruits Kib to the apical cortex, we asked if (1) Kib accumulates concurrently with aPKC at the apical cortex, (2) aPKC

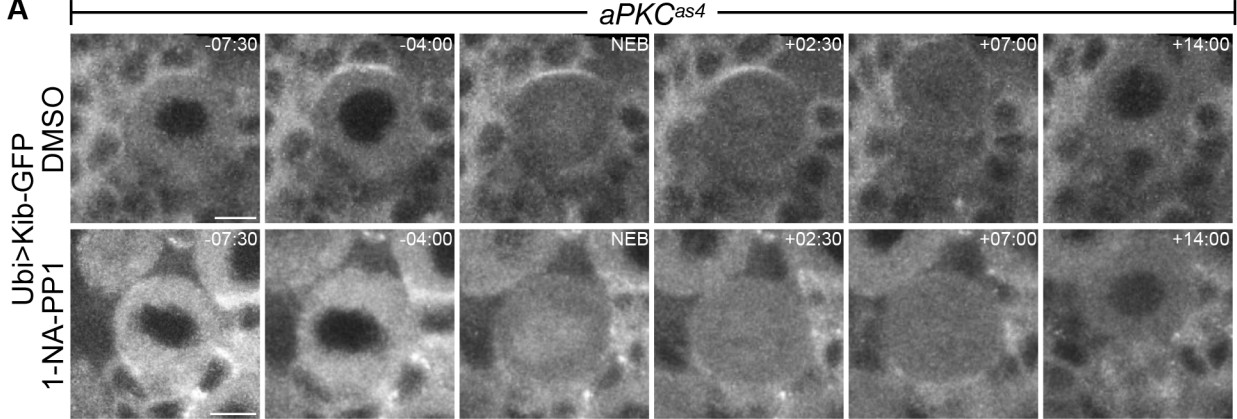

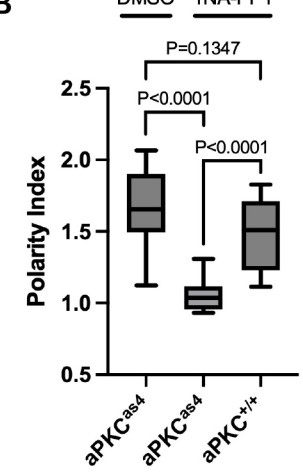

**Fig. 3. aPKC is required for Kibra polarization.** (A) Mitotic neuroblasts expressing Ubi>Kib-GFP from *aPKC^{as4}* animals treated with DMSO or the inhibitor 1-NA-PP1. (B) Quantification of the polarity index at NEB from multiple neuroblasts expressing Ubi>Kib-GFP from the experiment in A, as well as from *aPKC^{+/+}* animals treated with 1-NA-PP1 alone to control for its potential nonspecific effects. Data combined from two to three independent experiments (*aPKC^{as4}*, DMSO: n=9 neuroblasts; *aPKC^{as4}*, 1-NA-PP1: n=10 neuroblasts; *aPKC^{+/+}*, 1-NA-PP1: n=13 neuroblasts). Significance was determined by Student's unpaired *t*-test. Scale bars: 5 µm.

and Kib colocalize at the apical cortex, and (3) the aPKC-binding domain of Kib is necessary for Kib polarization.

To examine the dynamics of Kib accumulation in relation to aPKC, we quantified the rates of polarization of Ubi>Kib-GFP and aPKC-Halo in dividing neuroblasts. We found that their polarization dynamics differ in at least two significant ways. First, while Ubi>Kib-GFP and aPKC-Halo polarize at similar rates early in prophase, the rate of Ubi>Kib-GFP polarization peaks approximately 400 s before that of aPKC-Halo (Fig. 2E). Second, while aPKC-Halo continues to polarize after NEB, Ubi>Kib-GFP depolarizes immediately after NEB (Fig. 2D,E). The temporal difference in aPKC-Halo and Ubi>Kib-GFP polarization, in particular that Ubi>Kib-GFP polarization seems to precede that of aPKC-Halo, suggests that aPKC is unlikely to directly recruit Kib to the apical cortex.

To examine aPKC and Kib colocalization, we used Airyscan super-resolution imaging of the apical cortex of prophase neuroblasts expressing aPKC-Halo and Ubi>Kib-GFP. These images revealed that while both aPKC-Halo and Ubi>Kib-GFP accumulated at the apical cortex in puncta, many puncta were not colocalized and were instead adjacent to one another (Fig. 4A).

To test whether the aPKC-binding domain in Kib (Büther et al., 2004; Tokamov et al., 2023) is important for Kib polarization in neuroblasts, we observed the localization of a form of Kib lacking the aPKC-binding region (Ubi>Kib$^{\Delta aPKC}$-GFP) and a C-terminal fragment of Kib containing the aPKC-binding domain

(Ubi>Kib$^{858-1288}$-GFP). Interestingly, Ubi>Kib$^{\Delta aPKC}$-GFP polarized similarly to full-length Kib (Fig. 4B; Movie 7). Conversely, Ubi>Kib$^{858-1288}$-GFP failed to polarize (Fig. 4C; Movie 8). Therefore, the aPKC-binding region is neither necessary nor sufficient for Kib polarization in neuroblasts. Collectively, these results strongly suggest that aPKC does not promote Kib polarization by physically tethering it to the apical cell cortex.

## A pulsatile F-actin network promotes Kib polarization

In neuroblasts, apically directed actomyosin flows are necessary for proper aPKC polarization (Oon and Prehoda, 2019, 2021). Similarly, Kib localization is known to be regulated by actomyosin flows in epithelial cells (Tokamov et al., 2023). Therefore, we wondered if Kib could be polarized by actomyosin flows in neuroblasts.

To test this, we disrupted the F-actin network in live neuroblasts using Latrunculin A (Lat A) and examined the effect on Ubi>Kib-GFP polarization. Lat A-treated neuroblasts underwent karyokinesis but not cytokinesis, resulting in binucleate cells, confirming that Lat A treatment in these neuroblasts affected actin polymerization (Fig. 5B, yellow double-headed arrow). In control neuroblasts, aPKC-Halo and Ubi>Kib-GFP polarized as described previously (Fig. 5A,B; Movies 9 and 10). In Lat A-treated neuroblasts, aPKC-Halo polarized to the apical cortex prior to NEB (Fig. 5A; Movie 9). After NEB, however, aPKC-Halo spread toward the basal cortex rather than coalescing into a tight apical cap (Fig. 5A; Movie 9). These observations are consistent with previous observations of the

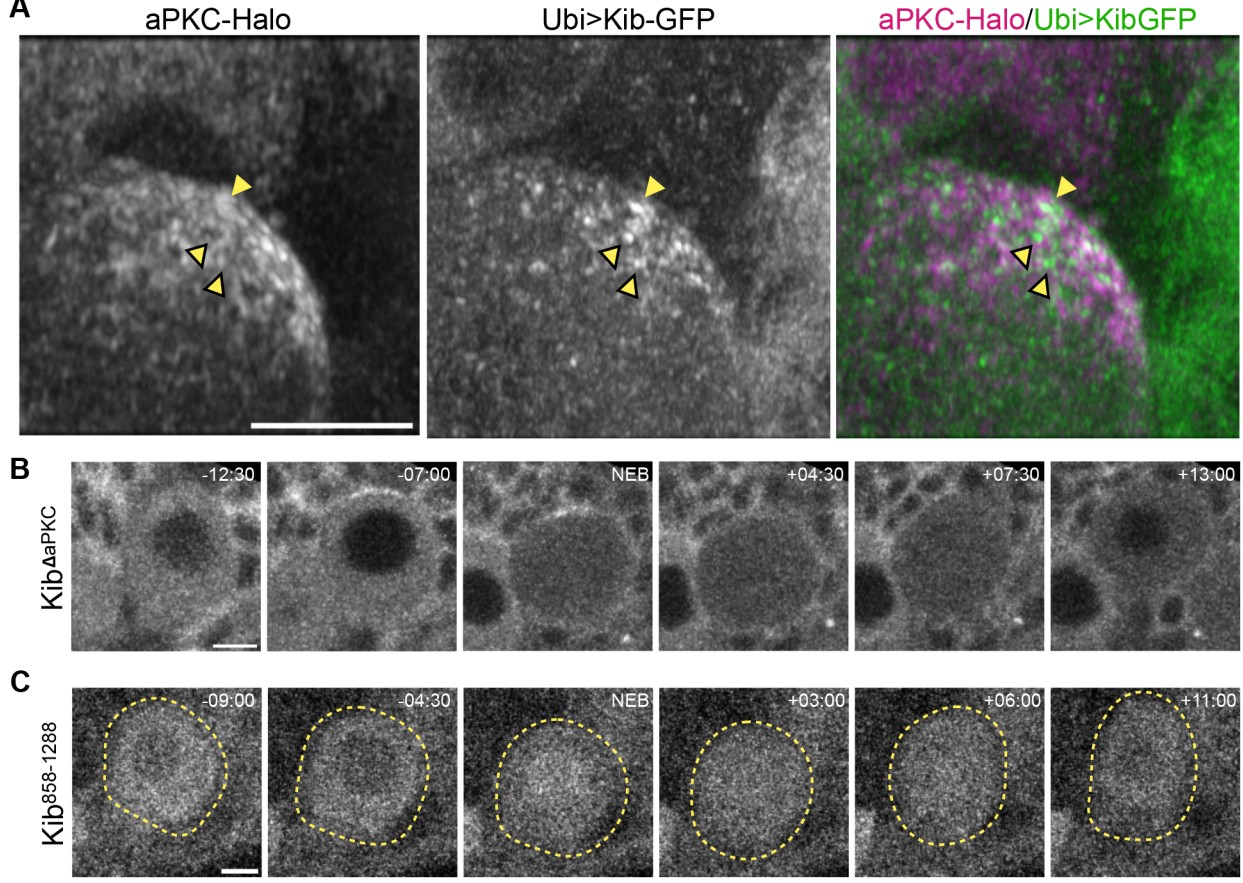

**Fig. 4. aPKC does not apically tether Kibra.** (A) A high-resolution view of the apical cortex of a neuroblast in prophase expressing aPKC-Halo and Ubi>Kib-GFP, using Airyscan. The plain yellow arrowhead shows colocalized aPKC-Halo and Ubi>Kib-GFP puncta, while the yellow arrowheads with black borders show the spatially separate puncta of aPKC-Halo and Ubi>Kib-GFP. (B) A mitotic neuroblast expressing Ubi>Kib$^{\Delta aPKC}$-GFP. (C) A mitotic neuroblast expressing Ubi>Kib$^{858-1288}$-GFP. The yellow dashed lines help demarcate the neuroblast. Scale bars: 5 µm.

Biology Open

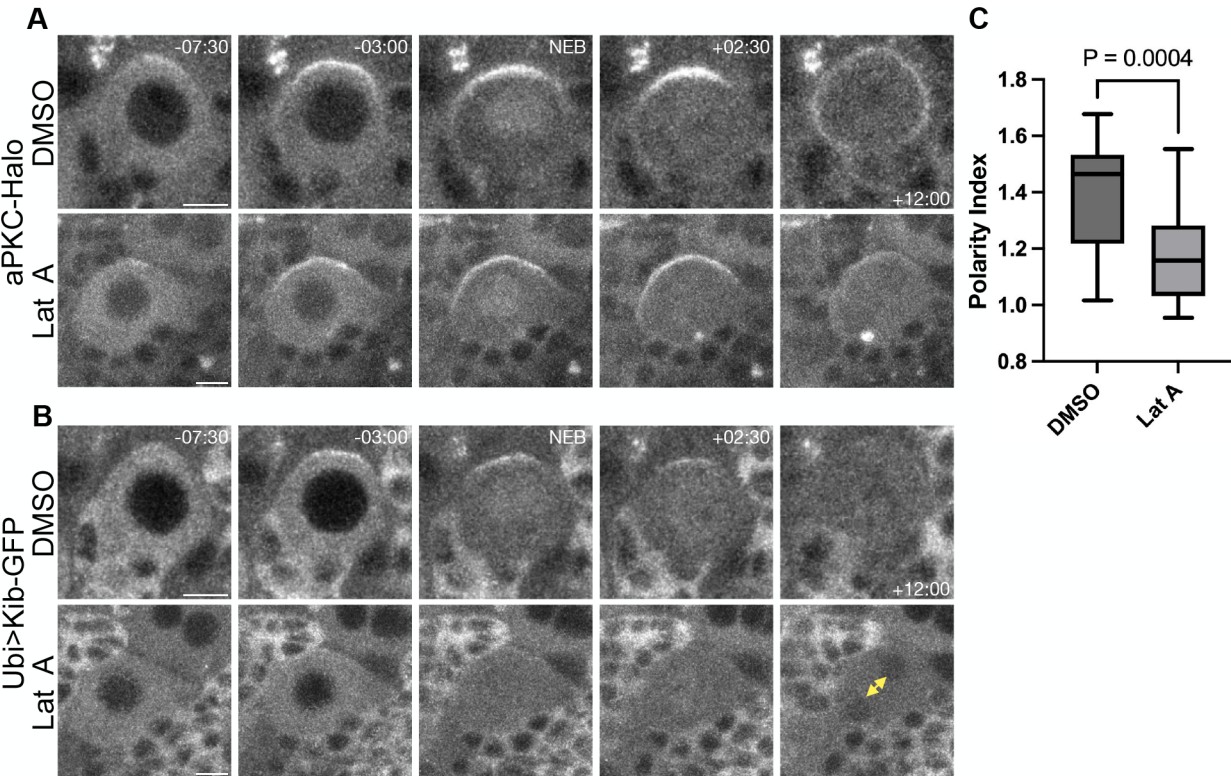

**Fig. 5. F-actin promotes Kib polarization.** (A) Polarization of aPKC-Halo in mitotic neuroblasts treated with DMSO or the F-actin inhibitor Lat A. (B) Polarization of Ubi>Kib-GFP in mitotic neuroblasts treated with DMSO or the F-actin inhibitor Lat A. The neuroblasts in this panel are the same as those in A. The yellow double-headed arrow points to the emergence of two nuclei in the Lat A-treated cell due to failed cytokinesis. (C) Quantification of the polarity index at NEB from multiple neuroblasts expressing Ubi>Kib-GFP from the experiment in A and B. Data combined from three independent experiments (DMSO: $n$=17 neuroblasts; Lat A: $n$=28 neuroblasts). Significance was determined by the Mann–Whitney $U$-test. Scale bars: 5 μm.

effect of Lat A on aPKC in larval brain neuroblasts (Oon and Prehoda, 2019).

Strikingly, Lat A-treated neuroblasts exhibited diminished and disorganized Ubi>Kib-GFP polarization (Fig. 5B; Movie 10). While we could observe some apical enrichment of Ubi>Kib-GFP just before and after NEB in Lat A-treated neuroblasts, Ubi>Kib-GFP failed to form strong and organized apical crescents in most neuroblasts (Fig. 5B; Movie 10), and the Ubi>Kib-GFP polarity index was lower in Lat A-treated neuroblasts than in controls (Fig. 5C). These results suggest that F-actin flows promote Kib apical polarization in dividing neuroblasts. Similar effects were observed using a different F-actin inhibitor, Cytochalasin D (CytoD), indicating that the effects of Lat A were specific to F-actin (Fig. S2).

In actively mitotic neuroblasts, cortical F-actin displays at least two different dynamics prior to telophase. During interphase, F-actin pulses form transient accumulations on the cell cortex, including on the apical cortex (Oon and Prehoda, 2021). Then, during prophase and around NEB, cortical F-actin flows towards the apical pole in a coordinated manner, where it accumulates stably until anaphase (Oon and Prehoda, 2021). Interphase F-actin pulses are thought to be required for initial stages of aPKC polarization, the inhibition of which results in a disorganized patchwork of aPKC on the apical cortex. Apically directed cortical F-actin flows before and around NEB are required for coalescence and maintenance of the aPKC apical crescents (Oon and Prehoda, 2019, 2021).

We wondered if F-actin pulses and apically directed flows contribute differently to Kib polarity. To examine the relationship between actin dynamics and Kib, we imaged Ubi>Kib-GFP with an F-actin marker, LifeAct-Halo, using Airyscan imaging. We were

able to image the entire cortex every 20 s at super-resolution, which provided a clear picture of Ubi>Kib-GFP and LifeAct-Halo dynamics. LifeAct-Halo displayed both pulsatile behavior during interphase and early prophase and stable apical accumulation around NEB (Fig. 6; Movie 11), as previously reported (Oon and Prehoda, 2021). Interestingly, Ubi>Kib-GFP accumulated at the apical cortex before NEB, when LifeAct-Halo dynamics were still pulsatile (Fig. 6A,B). After NEB, increasing apical accumulation of LifeAct-Halo corresponded with decreasing apical Ubi>Kib-GFP signal (Fig. 6C). Therefore, the timing of Kib polarization correlates more strongly with the period of F-actin pulsatile assembly than with its stable apical accumulation.

### Hippo pathway components downstream of Kib localize apically in larval neuroblasts

We next tried to determine how Hippo pathway components that function together with Kib localize in mitotic neuroblasts. Kib functions synergistically with Mer to assemble the core kinase complex (Su et al., 2017). Animals trans-heterozygous for *Mer* and *cno* display a highly penetrant ectopic neuron phenotype, which is indicative of compromised neuroblast polarity (Keder et al., 2015). Therefore, we wondered how Mer localizes in mitotic neuroblasts.

We visualized Mer by expressing the transgene Ubi>Mer-Halo in living neuroblasts. Ubi>Mer-Halo was present in the cytoplasm but strongly localized around the cell cortex in a symmetric manner (Movie 12). We did not observe dramatic Ubi>Mer-Halo redistribution prior to metaphase (Movie 12). However, Ubi>Mer-Halo flowed down from the apical cortex toward the basal cortex after metaphase, where it concentrated at the cytokinetic furrow (Movie 12).

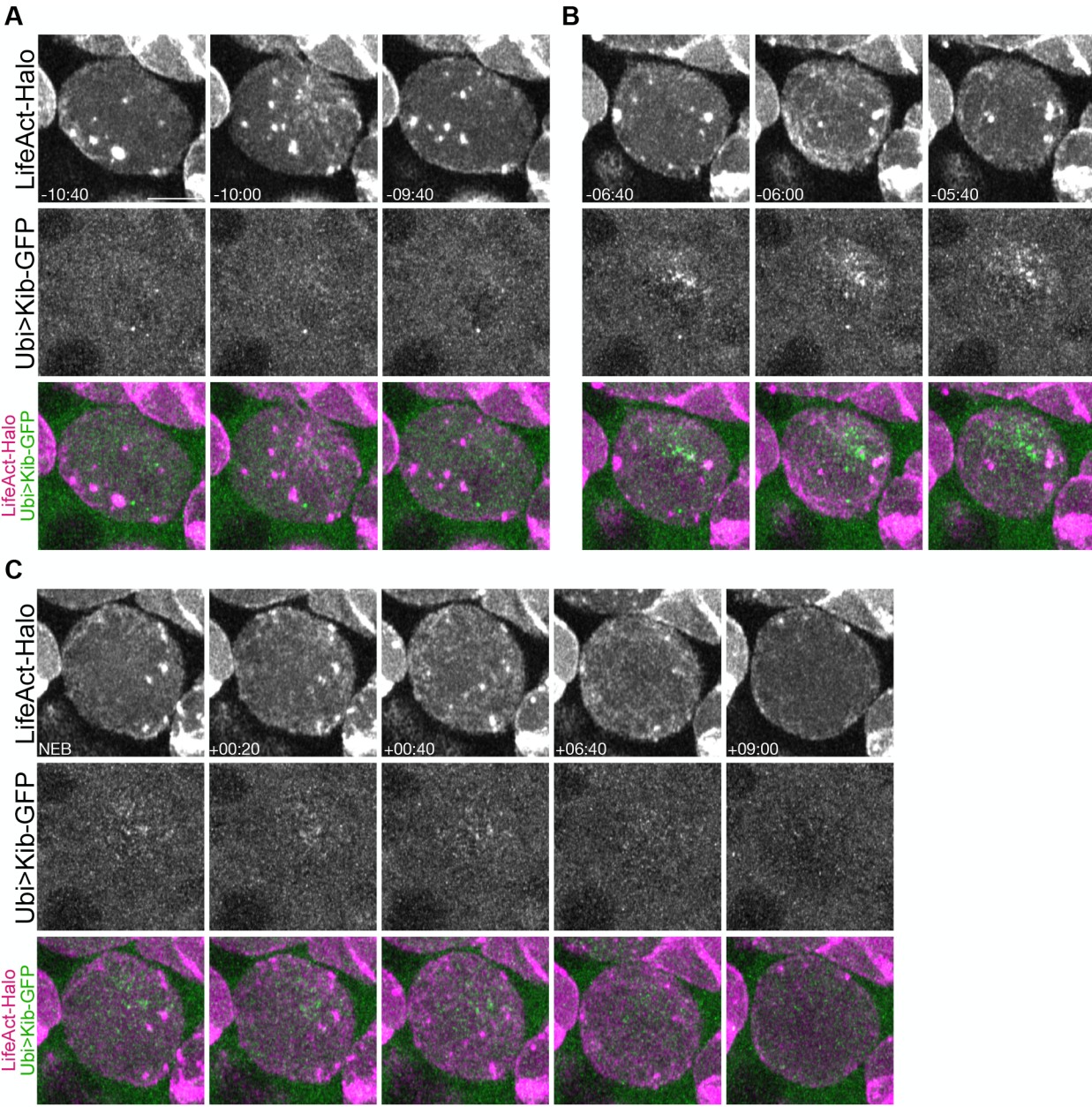

**Fig. 6. Kibra polarization is correlated with a pulsatile F-actin phase.** All panels display an apical projection of the same neuroblast expressing LifeAct-Halo and Ubi>Kib-GFP. (A) A pulse of F-actin on the apical cortex before Kib is polarized. (B) A pulse of F-actin on the apical cortex during Kib polarization. For A and B, the earliest panel displays the cortex before the pulse, the middle panel displays the cortex during the pulse, and the latest panel displays the cortex after the pulse. (C) The apical cortex during stable apical recruitment of F-actin. The earliest panel displays the cortex at the beginning of stable apical recruitment, which coincided with NEB in this neuroblast. The middle three panels display the cortex while F-actin remains stable apically. The latest panel displays the cortex as F-actin flows basally during telophase. Scale bar: 5 μm.

Kib and Mer activate the Hippo pathway by directly binding and recruiting the Hpo cofactor Sav (Yu et al., 2010; Su et al., 2017). Interestingly, loss of *sav* causes the most penetrant defect in neuroblast polarity of all Hippo pathway genes (Keder et al., 2015). Therefore, we wondered how Sav localizes in mitotic neuroblasts.

We visualized Sav localization by expressing the transgene Ubi>Sav-GFP in live neuroblasts. Strikingly, we found that Sav also polarizes during mitosis (Fig. 7A; Movie 13). Like Kib, Sav formed an apical crescent during prophase that diminished after NEB (Fig. 7A; Movie 13).

Because Kib interacts with Sav (Yu et al., 2010; Su et al., 2017), we wondered if they might polarize together in neuroblasts. To test

this possibility, we depleted Kib and Sav with previously validated RNAi transgenes (Su et al., 2017; Tokamov et al., 2021) expressed by a neuroblast-specific driver, *inscuteable>Gal4* (*insc>Gal4*). In control neuroblasts, Sav formed a bright apical crescent during prophase (Fig. 7A; Movie 13), which diminished after NEB, as described earlier. Contrary to our expectation, Sav still polarized when Kib was depleted (Fig. 7A; Movie 14). Conversely, Kib polarization was not perturbed by Sav depletion (Fig. 7B; Movie 15). Therefore, Kib and Sav polarize independently of each other in neuroblasts.

We attempted to elucidate which factors could promote Sav polarity by plotting its polarization dynamics (Fig. S3). On average,

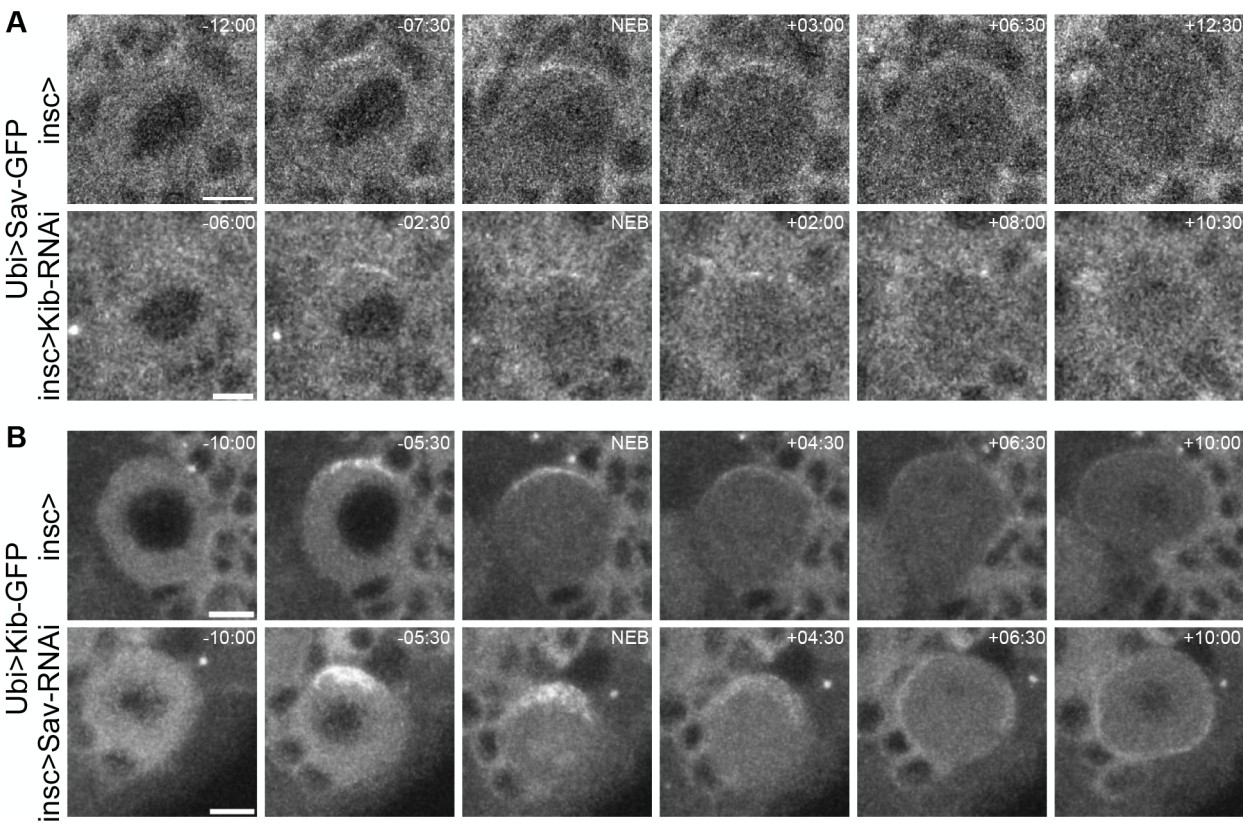

**Fig. 7. Salvador polarizes independently of Kibra.** (A) Top: a control mitotic neuroblast expressing Ubi>Sav-GFP shows weak polarization of Sav. Bottom: a mitotic neuroblast expressing Ubi>Sav-GFP in the background of Kib knockdown (insc>Kib-RNAi) – Sav is still polarized. (B) Top: a control mitotic neuroblast expressing Ubi>Kib-GFP. Bottom: a mitotic neuroblast expressing Ubi>Kib-GFP in the background of Sav knockdown (insc>Sav-RNAi) – Kib is still polarized. Scale bars: 5 µm.

Sav polarity peaks ~90 s after NEB, while aPKC, Baz, and Dlg are still polarizing, and Kib is already depolarizing (Fig. S3). Therefore, we could not identify a polarizing cue strongly correlated with Sav polarization.

### Wts does not polarize at the apical cortex

Finally, we examined the behavior of Wts in mitotic neuroblasts, where it is thought to be the key effector of Hippo signaling (Keder et al., 2015). To do so, we observed GFP-tagged Wts under its endogenous promoter (GFP:Wts) in living larval neuroblasts. We observed that Wts was predominantly diffusely cytoplasmic with discrete puncta in larval neuroblasts (Fig. 8A; Movie 16). Previous observations in fixed neuroblasts stained for Wts seem to show that Wts becomes apically polarized in the cytoplasm in metaphase (Keder et al., 2015). To describe Wts localization dynamics in living neuroblasts, we calculated the mean ratio between GFP:Wts intensity at the apical cytoplasm and basal cytoplasm (cytoplasmic polarity index) over time in multiple cycling neuroblasts (Fig. 8B). Minutes before NEB, we found that the apical cytoplasm was slightly enriched with GFP:Wts. However, GFP:Wts became slightly less enriched in the apical cytoplasm after NEB. These observations suggest that Wts localization does not change dramatically during asymmetric cell division in neuroblasts.

### DISCUSSION

Genetic data have implicated the Hippo pathway as a regulator of cortical polarity and spindle orientation in asymmetrically dividing neuroblasts (Dewey et al., 2015; Keder et al., 2015). However, the mechanism of Hippo pathway activation and the organization of its

upstream activators in this context remain unknown. In this study, we characterized the localization kinetics of key Hippo signaling proteins throughout neuroblast division. Surprisingly, we found that the Hippo pathway activator Kib and core complex component Sav polarize to the apical cortex during mitosis in a manner that is reminiscent of, but distinct from, the polarization of the Par proteins.

Our experiments show that Kib polarization is linked to the activity of the apical complex kinase aPKC. Kib's initial polarization during early prophase proceeds at a similar rate as that of aPKC and requires aPKC activity. Several lines of evidence, however, suggest that the apical Par complex does not tether Kib to the apical cortex. First, the dynamics of Kib polarization and depolarization do not resemble that of aPKC and Baz after early prophase. Second, apical Kib and aPKC puncta often do not colocalize. Third, a known Kib-aPKC interaction domain is neither necessary nor sufficient for Kib's polarization. However, our data do not rule out the possibility that Kib transiently interacts with the Par proteins at the apical cortex. This interaction potentially could modify Kib activity through its phosphorylation by aPKC, or it could inhibit Par complex activity as has been shown in other contexts (Yoshihama et al., 2011, 2012; Jin et al., 2015).

Our experiments uncovered that another major positive regulator of Kib polarization is F-actin. We found that inhibiting F-actin organization with Lat A diminished Kib polarization. Previous work has shown that two different F-actin dynamics – transient pulses during interphase and apically directed flow during prophase and at NEB – contribute distinctly to aPKC and Baz polarization: While interphase pulses are required for initially recruiting aPKC and Baz to the cell cortex, apically directed flows cause coalescence of a tight crescent at the apical cortex (Oon and Prehoda, 2019, 2021).

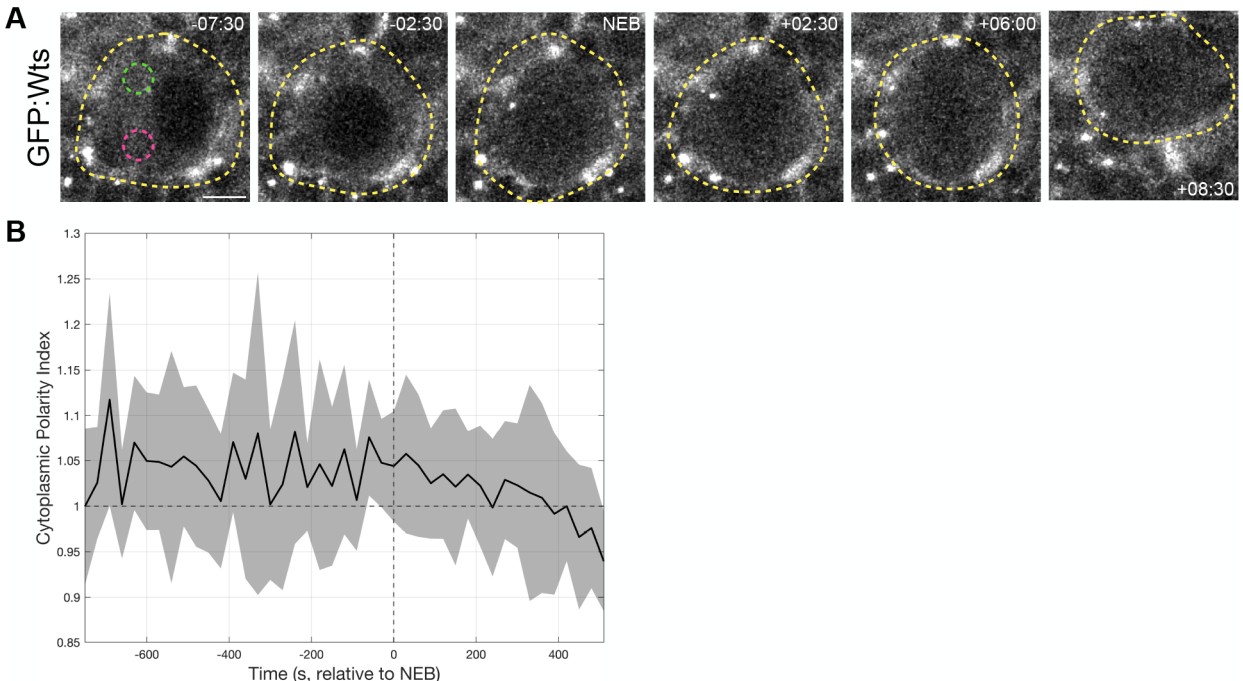

**Fig. 8. Warts does not polarize at the apical cortex.** (A) A mitotic neuroblast expressing endogenously expressed GFP:Wts. The yellow dashed line helps demarcate the neuroblast. Green and magenta dashed line circles represent apical and basal cytoplasm areas, respectively, used to quantify the ratios in B. (B) Quantification of the ratio between the intensity of apical cytoplasmic GFP:Wts and basal cytoplasmic GFP:Wts (cytoplasmic polarity index). Eight neuroblasts from two independent experiments are represented by the curve, and each timepoint is the average of at least three neuroblasts. Shading represents one standard deviation from the mean. The vertical dotted line marks time=NEB, and the horizontal dashed line marks the cytoplasmic polarity index=1. Scale bar: 5 μm.

Interestingly, Kib polarization occurs while F-actin is pulsatile, and Kib depolarizes just after NEB, when F-actin flows are still concentrating the aPKC/Baz crescent. To the extent that proteins rely on F-actin to polarize in neuroblasts, this suggests that the outcome of these two F-actin dynamics differs from protein to protein. It may also suggest that some form of regulation sensitizes Kib to interphase F-actin pulses but not apically directed flow during mitosis.

How do the Par complex and F-actin synergize to promote Kib polarization? One possibility is that aPKC phosphorylates Kib to increase its sensitivity to cortical F-actin flow. Indeed, Kib and its orthologs are regulated by phosphorylation (Büther et al., 2004; Tokamov et al., 2021), including in a cell cycle-dependent manner (Xiao et al., 2011). Phosphorylation itself might increase Kib's affinity for F-actin, as it does for other proteins (Nakamura et al., 1999; Janji et al., 2006; Bosk et al., 2011) or promote Kib's interaction with an intermediate that itself is carried by F-actin flow.

The observation that Kib polarization depends on both aPKC activity and F-actin suggests an additional possibility – that aPKC promotes Kib apical polarization by affecting actomyosin dynamics. Previous studies have shown that aPKC modulates actomyosin contractility and flow in a variety of polarized contexts, including the *Caenorhabditis elegans* zygote (Packer et al., 2024 preprint) and epithelia (Kishikawa et al., 2008; David et al., 2013; Biehler et al., 2021; Osswald et al., 2022). In particular, aPKC seems to antagonize actomyosin contractility (Kishikawa et al., 2008; David et al., 2013; Osswald et al., 2022). Consistent with this, we observed that aPKC inhibition causes membrane blebbing at the neuroblast cortex (Fig. S4; Movie 17), a hallmark of increased actomyosin contractility (García-Arcos et al., 2024). Therefore, it is possible that aPKC-mediated regulation of actomyosin contractility is necessary for the cortical actomyosin dynamics that polarize Kib. The results presented

here do not fully explain how aPKC and actomyosin together promote Kib polarization. Further work will be required to elucidate the complex relationship between aPKC and actomyosin contractile networks in dividing neuroblasts.

In the wing imaginal disc cells, Kib distribution at the apical cortex is regulated by the opposing action of aPKC and F-actin (Tokamov et al., 2023), whereby aPKC tethers Kib at cell junctions and F-actin flows sweep Kib to the medial cortex (Tokamov et al., 2023). Loss of aPKC causes concentration of Kib at the medial cortex because of a loss of junctional tethering and greater medial F-actin flows (Tokamov et al., 2023). In neuroblasts, however, aPKC inhibition leads to Kib being diffuse in the cytoplasm and to lower accumulation of apical F-actin (Fig. S4, Movie 17). Together, this suggests that aPKC activity and F-actin flows generally function as regulators of Kib localization across contexts. However, the fact that changes to F-actin flow and Kib localization upon aPKC inhibition differ between these tissues demonstrates that additional context-dependent regulation is involved.

What is the significance of the polarized localization of Kib we describe here? The requirement for both upstream activators and members of the Hippo pathway core kinase cassette for asymmetric neuroblast division suggests that the ability of Wts to phosphorylate apical targets like Baz, Cno, and Mud depends on its activation in a pathway-dependent manner (Dewey et al., 2015; Keder et al., 2015). Further, the observations that Wts is weakly enriched apically in dividing neuroblasts and its targets – Baz, Cno and Mud – are simultaneously apically polarized (Keder et al., 2015) suggest that Wts is likely preferentially activated at the apical cortex. In epithelial cells, Kib, in concert with Mer and Sav, has been shown to recruit both Hpo and Wts to the medial apical cortex and thereby activate Hippo signaling (Su et al., 2017; Tokamov et al., 2023). The results presented here reveal that Kib and Sav undergo dramatic apical

Biology Open

polarization in concert with the polarization of neuroblasts during mitosis and that Mer is simultaneously present on the apical cortex. Given these observations, we speculate that in mitotic neuroblasts, apical accumulation of Kib and Sav could result in local activation of Wts in the apical domain and, as a result, phosphorylation of apically localized Cno and Mud. An important part of this model is that it suggests that Hippo pathway activity in neuroblasts may be subject to interesting forms of regulation, such as positive feedback with the Par complex. Unfortunately, the weak penetrance of spindle orientation defects in Hippo pathway mutants (Keder et al., 2015), coupled with the current lack of biosensors for Wts activity (or phospho-specific antibodies that detect endogenous levels of Wts expression), prevented us from testing this model directly. Future work should interrogate the consequence of Kib depletion on neuroblast polarity and continue to explore functional interactions between apical polarity and Hippo signaling, which so far have not been thoroughly examined.

## MATERIALS AND METHODS
### Reagents and stocks
Information on chemical reagents and *Drosophila* stocks can be found in Tables S1 and S2.

### *Drosophila* husbandry
*Drosophila melanogaster* was cultured using standard techniques at 25°C. Male and female larvae were used for brain dissections.

### Live imaging
Brain explants were dissected from non-wandering third-instar larvae (i.e. 72-96 h after egg laying) in Schneider's *Drosophila* medium supplemented with 5% fetal calf serum (FCS) (Sigma-Aldrich, 'dissection medium') on a siliconized glass slide. Brains significantly damaged during the process of dissection were discarded. Three brains were mounted at a time in an imaging chamber (Restrepo et al., 2016) to image live *Drosophila* wing imaginal discs. Brains were mounted with the ventral side facing the coverslip. Only neuroblasts within the brain lobes, not the ventral nerve cord, were imaged.

All images were taken using either an oil-immersion 40× PlanApo (1.4 NA) or a water-immersion 40× c-Apo (1.2 NA) objective on an inverted Zeiss LSM 880 or LSM 980 confocal microscope equipped with a GaAsP spectral detector. Airyscan Fast was used where noted in the text and figure legends.

### Fluorescent labeling and chemical genetics
For Halo labeling, three brain explants were incubated in 100 µl of a solution of Halo dye diluted in dissection medium or Ringers on a siliconized glass slide in a humid chamber for 15 min. Either a 1 µM solution of Halo dye JFX-650 or a 0.5 µM solution of Halo dye JF-646 was used. Explants were then washed in 100 µl of dissection medium on a siliconized glass slide in a humid chamber for 5 min. Brains were then mounted for live imaging.

For drug treatment, three brains were incubated at a time in a drug solution (1-NA-PP1, 10 µM; Lat A 50 µM) diluted in dissection medium on a siliconized glass slide in a humid chamber for 15 min prior to mounting and imaging, either simultaneous with or subsequent to Halo labeling. For CytoD treatment, brains were incubated in a 100 µM solution for 1 h. Drug concentrations were based on existing protocols (Hannaford et al., 2019; LaFoya and Prehoda, 2021). For control experiments, an equivalent volume of DMSO diluted in dissection medium was used. Brains were mounted for imaging in the drug or DMSO solutions.

### Quantification
Mean apical crescent intensity was measured by selecting an area over approximately the middle half of the apical cortex with the segmented line tool in Fiji/ImageJ. The line width was set to 3. Cytoplasmic fluorescence intensity was determined by measuring the mean fluorescence intensity of a circular section of the cytoplasm using the elliptical tool in Fiji. The diameter of the circle was set to 15-20 pixels. Basal cytoplasmic

intensity was measured close to the basal cortex to minimize detection of apical signal while also avoiding overlap with the nucleus. Apical cytoplasm measurements avoided overlap with the apical cortical signal. Measurements were performed on maximum intensity projections of four optical sections on raw images. Any neuroblasts in which the apical crescent could not be viewed in at least four optical sections or those that divided significantly out of the imaging plane were not quantified.

Time dynamics were plotted using MATLAB. The 'gradient' function was used to compute derivatives of polarization curves. Derivatives were smoothed using a Gaussian filter with the 'smoothdata' function.

### Statistical analysis
Multiple neuroblasts across multiple independent larval brains were analyzed for each experiment. At least two biological replicates (i.e. experiments done on different days) were performed for each result. No predetermined sample size was calculated.

Data were tested for normality using the Shapiro–Wilk test. If the data were normal, significance between experimental conditions was determined using Student's *t*-test. If the data were not normal, significance between experimental conditions was determined using the Mann–Whitney *U*-test. Tests for significance were two-tailed, and data were considered significant if $P<0.05$. All statistical analyses were done using GraphPad Prism.

### Acknowledgements
Stocks obtained from the Bloomington *Drosophila* Stock Center (NIH P40OD018537) were used in this study. The JFX-650 Halo dye was provided by the Lavis laboratory at the Janelia Research Campus. We thank the members of the Fehon laboratory for helpful discussions.

### Competing interests
The authors declare no competing or financial interests.

### Author contributions
Conceptualization: N.S.J., V.M.S., S.A.T., R.G.F.; Formal analysis: N.S.J., V.M.S.; Funding acquisition: R.G.F.; Investigation: N.S.J., V.M.S., S.A.T.; Supervision: S.A.T., R.G.F.; Writing – original draft: N.S.J., R.G.F.; Writing – review & editing: N.S.J., V.M.S., S.A.T., R.G.F.

### Funding
N.S.J. was supported by an Academic Year Fellowship, a Developmental Neurobiology Undergraduate Fellowship, a Genetics and Genomics Undergraduate Fellowship, a Quad Undergraduate Research Scholarship, and funding from the Research Honors Program from the University of Chicago. V.M.S. was supported by funding from the Jeff Metcalf Internship Program and a Research Foundations in Genetics and Genomics Undergraduate Fellowship from the University of Chicago. S.A.T. was supported by the National Institute of General Medical Sciences (GM007183) and the National Science Foundation Graduate Research Fellowship Program. This work was supported by a grant from the National Institutes of Health awarded to R.G.F. (R01NS034783). Open Access funding provided by University of Chicago. Deposited in PMC for immediate release.

### Data and resource availability
All relevant data and details of resources can be found within the article and its supplementary information. Data are also available at Zenodo (https://zenodo.org/records/18355118).

### First Person
This article has an associated First Person interview with the first author of the paper.

### Peer review history
The peer review history is available online at https://journals.biologists.com/bio/lookup/doi/10.1242/bio.062356.reviewer-comments.pdf

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
