## [Peer Review File · Biology Open]

aPKC and F-actin Dynamics Promote Hippo Pathway Polarity in Asymmetrically Dividing Neuroblasts

Niranjan S. Joshi, Victoria M. Sullivan, Sherzod A. Tokamov and Richard G. Fehon
DOI: 10.1242/bio.062356

Editor: Sandhya Koushika

Review timeline

Original submission:	4 November 2025
Editorial decision:	14 November 2025
First revision received:	27 January 2026
Accepted:	28 January 2026

Original submission

First decision letter

MS ID#: bio.062356

MS Title: aPKC and F-actin Dynamics Promote Hippo Pathway Polarity in Asymmetrically Dividing Neuroblasts

Authors: Niranjan S. Joshi, Victoria M. Sullivan, Sherzod A. Tokamov and Richard G. Fehon

I have now reached a decision on the above manuscript.

The reviewer reports are shown at the bottom of this email.

As you will see, the reviewers gave favourable reports, but raised some critical points that will require amendments to your manuscript. I hope that you will be able to carry these out, because we would like to be able to accept your paper.

Reviewer 1

Comments for the author

The research investigates how the Hippo signaling pathway—a key regulator of growth and polarity—is spatially organized during asymmetric cell division in *Drosophila* neuroblasts. The authors focus on two Hippo activators, Kibra (Kib) and Salvador (Sav), demonstrating that both proteins polarize apically in mitotic neuroblasts. They further show that aPKC kinase activity and F-actin dynamics cooperate to drive this polarization, providing new mechanistic insight into how Hippo signaling integrates with apical-basal polarity cues.

The use of live imaging in intact larval brains and endogenous or minimally overexpressed fluorescent tags (e.g., Kib::YFP, Ubi>Kib-Halo) provides convincing spatiotemporal data. The aPKCas4 allele combined with acute 1-NA-PP1 inhibition allows precise temporal control of aPKC activity—a strong approach to dissect kinase dependency. Use of Latrunculin A for F-actin perturbation is well controlled and consistent with prior neuroblast studies.

I have some questions which authors can address :

1) While Kib polarization depends on aPKC activity and actin integrity, the causal biochemical link remains unclear. The data exclude direct tethering but do not identify the intermediate mechanism or phosphorylation targets.

2) Incomplete Analysis of Downstream Pathway: Although Warts (Wts) localization is examined, its activation state (e.g., phosphorylation status) is not assayed. This limits insight into how polarization translates into differential Hippo pathway activity.

3) The independence of Kib and Sav polarization is intriguing but could benefit from colocalization or cross-correlation analyses to quantify temporal overlap.

Reviewer 2

Comments for the author

This manuscript by Joshi et al. reports that two key activators of the Hippo pathway, Kibra and Salvador, polarize to the apical cortex of mitotic neuroblasts through distinct mechanisms. Kibra polarization requires both aPKC kinase activity and pulsatile F-actin dynamics, indicating that apical polarity and cytoskeletal organization synergize to drive its localization. The authors propose that these findings establish a mechanistic link between apical polarity, the F-actin cytoskeleton, and Hippo pathway regulation, providing insight into how pathway activity is controlled during neuroblast asymmetric division.

The paper is interesting, well executed, and includes appropriate experimental controls. Particularly, the use of live imaging adds significant value.

However, I find the study makes claims that are not completely supported by the data. My comments below, try to address this.

This study does not directly measure Hippo pathway activity or downstream functional outputs (for instance Warts kinase activity, substrate phosphorylation, spindle orientation, or fate determinant segregation) yet concludes that aPKC and F-actin dynamics promote pathway polarity (Figures 3-8). Without functional readouts, inferring pathway regulation solely from protein localization remains speculative. This limitation weakens the claim that Kibra and Salvador dynamics actively regulate Hippo pathway activity during neuroblast asymmetric division. If pathway activity is not assessed, the manuscript should revise its language to acknowledge that this interpretation is speculative and clarify that the work primarily establishes the spatial organization of Hippo pathway components in neuroblasts, rather than their activity.

The conclusion that aPKC activity is required for Kibra polarization is based on acute inhibition using 1-NA-PP1 in the aPKC^{as4} background, where DMSO-treated neuroblasts display strong apical Kibra crescents, while 1-NA-PP1-treated cells do not (Figure 3A-B). This chemical-genetic strategy is a powerful and widely accepted approach for probing kinase function in a time-resolved manner. The imaging and quantification convincingly demonstrate a loss of Kibra polarization upon inhibitor treatment. However, it lacks specificity controls and/or genetic rescue experiments. For example, the authors should include a wild-type control plus 1-NA-PP1, dose-response analysis, or rescue with an inhibitor-resistant aPKC variant to rule out potential off-target effects of 1-NA-PP1 in the Kibra polarization assay.

The study's reliance on a single actin-depolymerizing compound at one dose, without a washout/recovery control (or a genetic control), raises concerns about specificity. Using only Latrunculin A (Figure 5A-C) leaves open the possibility of compound-specific or off-target effects, especially given observed changes in aPKC. Standard actin research protocols include dose-response and reversibility tests to confirm specificity. Without these, it is difficult to conclude that F-actin integrity itself is required rather than a Latrunculin A-specific effect.

Reviewer's Responses to Questions

Experimental quality

Does each figure have the proper controls?

If 'No', please indicate reasons in Comments for Author box below.

Reviewer #1:

- Yes

Reviewer #2:

- No
-

Were the data analyzed using appropriate statistical tests?

If 'No', please indicate reasons in Comments for Author box below.

Reviewer #1:

- Yes

Reviewer #2:

- Yes
-

Reproducibility

Were experiments performed using adequate number of biological replicates?

If 'No', please indicate reasons in Comments for Author box below.

Reviewer #1:

- Yes

Reviewer #2:

- Yes
-

Does the methods section provide sufficient detail to permit reproducibility?

If 'No', please indicate reasons in Comments for Author box below.

Reviewer #1:

- Yes

Reviewer #2:

- Yes
-

Completeness

Are the manuscript's conclusions supported by the data?

If 'No', please indicate reasons in Comments for Author box below.

Reviewer #1:

- Yes

Reviewer #2:

- Yes
-

Scholarship

Do the authors cite and discuss the merits of data that would argue for and against their conclusion?

If 'No', please indicate reasons in Comments for Author box below.

Reviewer #1:

- Yes

Reviewer #2:

- Yes
-

Does the manuscript title & abstract accurately reflect the contents of the manuscript, without hyperbole?

If 'No', please indicate reasons in Comments for Author box below.

Reviewer #1:

- Yes

Reviewer #2:

- No

First revision

Author response to reviewers' comments

We thank both reviewers for their thoughtful comments and were pleased to see that they felt this work is of high quality and significant interest. In the revised version we have addressed these comments both through modification of the text and addition of new data. We think these changes address the concerns that were raised and hope the reviewers will agree.

Reviewer 1: The research investigates how the Hippo signaling pathway—a key regulator of growth and polarity—is spatially organized during asymmetric cell division in Drosophila neuroblasts. The authors focus on two Hippo activators, Kibra (Kib) and Salvador (Sav), demonstrating that both proteins polarize apically in mitotic neuroblasts. They further show that aPKC kinase activity and F-actin dynamics cooperate to drive this polarization, providing new mechanistic insight into how Hippo signaling integrates with apical-basal polarity cues.

The use of live imaging in intact larval brains and endogenous or minimally overexpressed fluorescent tags (e.g., Kib::YFP, Ubi>Kib-Halo) provides convincing spatiotemporal data. The aPKC^{as4} allele combined with acute 1-NA-PP1 inhibition allows precise temporal control of aPKC activity—a strong approach to dissect kinase dependency. Use of Latrunculin A for F-actin perturbation is well controlled and consistent with prior neuroblast studies.

I have some questions which authors can address :

1) While Kib polarization depends on aPKC activity and actin integrity, the causal biochemical link remains unclear. The data exclude direct tethering but do not identify the intermediate mechanism or phosphorylation targets.

We agree with the reviewer that although our data clearly implicate aPKC activity and actin integrity in Kibra polarization, the exact biochemical link remains unclear. We have added a statement in the discussion to emphasize this point.

2) Incomplete Analysis of Downstream Pathway: Although Warts (Wts) localization is examined, its activation state (e.g., phosphorylation status) is not assayed. This limits insight into how polarization translates into differential Hippo pathway activity.

We agree that direct analysis of Warts activity in the apical cortex would be interesting, but unfortunately the necessary tools for this analysis do not exist. Antibodies specific for phospho-Warts have been generated by other labs but in our hands they are not sufficiently sensitive to detect endogenously-expressed Warts in tissue stains. We have added statements at the end of the discussion to make clear that this is a speculative model that will require further study to fully test.

3) The independence of Kib and Sav polarization is intriguing but could benefit from colocalization or cross-correlation analyses to quantify temporal overlap.

We thank the reviewer for this suggestion. This is a good idea, with the caveat that the signal to noise is low for Sav due to its low expression level making any quantitative analysis difficult. We plotted the polarity index during mitosis for Sav and present this in Fig. S3, together with the plots for Kib and aPKC from Fig. 2 for comparison, but do not think that we can compare more quantitatively. Although the peak of Sav polarization is temporally a bit later than that of Kib (and earlier than aPKC), we don't think that these results clearly distinguish Sav temporal dynamics from those of Kib.

Reviewer 2:

This manuscript by Joshi et al. reports that two key activators of the Hippo pathway, Kibra and Salvador, polarize to the apical cortex of mitotic neuroblasts through distinct mechanisms. Kibra polarization requires both aPKC kinase activity and pulsatile F-actin dynamics, indicating that apical polarity and cytoskeletal organization synergize to drive its localization. The authors propose that these findings establish a mechanistic link between apical polarity, the F-actin cytoskeleton, and Hippo pathway regulation, providing insight into how pathway activity is controlled during neuroblast asymmetric division.

The paper is interesting, well executed, and includes appropriate experimental controls. Particularly, the use of live imaging adds significant value.

However, I find the study makes claims that are not completely supported by the data. My comments below, try to address this.

This study does not directly measure Hippo pathway activity or downstream functional outputs (for instance Warts kinase activity, substrate phosphorylation, spindle orientation, or fate determinant segregation) yet concludes that aPKC and F-actin dynamics promote pathway polarity (Figures 3-8). Without functional readouts, inferring pathway regulation solely from protein localization remains speculative. This limitation weakens the claim that Kibra and Salvador dynamics actively regulate Hippo pathway activity during neuroblast asymmetric division. If pathway activity is not assessed, the manuscript should revise its language to acknowledge that this interpretation is speculative and clarify that the work primarily establishes the spatial organization of Hippo pathway components in neuroblasts, rather than their activity.

We agree with the reviewer. As noted above, we have added statements at the end of the discussion to state explicitly the limitations of the experiments we have been able to perform and that the model we proposed is speculative.

The conclusion that aPKC activity is required for Kibra polarization is based on acute inhibition using 1-NA-PP1 in the aPKCas4 background, where DMSO-treated neuroblasts display strong apical Kibra crescents, while 1-NA-PP1-treated cells do not (Figure 3A-B). This chemical-genetic strategy is a powerful and widely accepted approach for probing kinase function in a time-resolved manner. The imaging and quantification convincingly demonstrate a loss of Kibra polarization upon inhibitor treatment. However, it lacks specificity controls and/or genetic rescue experiments. For example, the authors should include a wild-type control plus 1-NA-PP1, dose-response analysis, or rescue with an inhibitor-resistant aPKC variant to rule out potential off-target effects of 1-NA-PP1 in the Kibra polarization assay.

We thank the reviewer for this suggestion. Although we had shown previously that 1-NA-PP1 does not affect aPKC activity in wild-type wing discs (Tokamov et al. 2023), we had not included this control for larval brains. This control is now included in Figure 3B.

The study's reliance on a single actin-depolymerizing compound at one dose, without a washout/recovery control (or a genetic control), raises concerns about specificity. Using only Latrunculin A (Figure 5A-C) leaves open the possibility of compound-specific or off-target effects, especially given observed changes in aPKC. Standard actin research protocols include dose-response and reversibility tests to confirm specificity. Without these, it is difficult to conclude that F-actin integrity itself is required rather than a Latrunculin A-specific effect.

The reviewer raises an important point. We would first note that previous work from the Prehoda lab, whose protocols we followed, has demonstrated the specificity of LatA in this context. In

addition, we have added new data to address this concern. We found that a wash out approach was not practical in this context due to the inability to maintain brain explants in culture long enough for recovery to occur. As an alternative, we tested the effect of Cytochalasin D, which the Prehoda lab also used previously on larval brains. We found that similar to LatA, Cytochalasin D blocked Kibra polarization in mitotic neuroblasts. These data are presented in a new Figure S2.

Second decision letter

MS ID#: bio.062356R1

MS Title: aPKC and F-actin Dynamics Promote Hippo Pathway Polarity in Asymmetrically Dividing Neuroblasts

Authors: Niranjan S. Joshi, Victoria M. Sullivan, Sherzod A. Tokamov and Richard G. Fehon

I am happy to tell you that your manuscript has been accepted for publication in Biology Open, pending our standard publication integrity checks. It was accepted on 28th January 2026.